# Study of the Preparation and Performance of TiO_2_-Based Magnetic Regenerative Adsorbent

**DOI:** 10.3390/molecules29132964

**Published:** 2024-06-21

**Authors:** Dahui Tian, Jiarui Hao, Xiaojie Guo

**Affiliations:** Department of Materials and Chemical Engineering, Taiyuan University, Taiyuan 030024, China; tiandahui@tyu.com (D.T.);

**Keywords:** TiO_2_, γ-Fe_2_O_3_ coal-series kaolin, renewable materials, adsorption, Congo red

## Abstract

Against the backdrop of “carbon neutrality”, the green treatment of dye wastewater is particularly important. Currently, the adsorption method shows strong application prospects. Therefore, selecting efficient and recyclable adsorbents is of significant importance. TiO_2_ is an excellent adsorbent, but its difficult recovery often leads to secondary pollution. γ-Fe_2_O_3_-modified coal-series kaolin exhibits both excellent adsorption properties and rapid separation through magnetic separation technology. By utilizing the synergistic effects of both, TiO_2_/-γFe_2_O_3_ coal-series kaolin, magnetic adsorbent regeneration materials were prepared using coprecipitation method and characterized. The influencing factors of this functional material on the adsorption of Congo red dye and its regeneration performance are discussed. The experimental results indicated that the specific surface area, pore volume and Ms value of this functional material are 127.5 m^2^/g, 0.38 cm^3^/g, and 13.4 emu/g, respectively. It exhibits excellent adsorption characteristics towards Congo red, with an adsorption rate reaching 96.8% within 10 min, conforming to the pseudo-second-order kinetic model, and demonstrating Langmuir IV-type monolayer adsorption. After the adsorption of Congo red, magnetic separation shows superior efficiency. Furthermore, treatment of the adsorbed composite with EDTA allows for recycling, with adsorption rates still above 91% after three cycles, indicating an excellent regeneration capability.

## 1. Introduction

As widely used colorants, dyes are extensively employed in various fields such as food, medicine, leather making, paint, plastics, optoelectronics and communications [1]. Currently, the global annual production of dyes is approximately 1.2 million tons, with China being the world’s largest exporter, accounting for over 70% of the total production [2]. Dyes come in a variety of types and involve complex processes [3,4]. During synthesis and usage, approximately 10–15% of dyes are discharged into the environment [5]. Most dye wastewater exhibits stable properties, complex compositions, high chromaticity toxicity, poor biodegradability and high concentrations of chemical oxygen demand (COD) [6]. Once released into the environment, these dyes cannot undergo natural degradation [7], leading to environmental pollution and exerting a triple effect [8]. Consequently, the treatment of dye wastewater has become a globally recognized issue.

Dye wastewater varies in composition, leading to various treatment methods. Based on removal mechanisms, the treatment methods mainly fall into two categories: separation and degradation. Among them, commonly used separation methods include coagulation, membrane separation, adsorption, magnetic separation and so on. Coagulation is the most widely applied due to its simplicity, low cost and effective adsorption [9,10]. However, it is less effective when dealing with acidic, basic, cationic, azo and reactive dyes with relatively small molecular weights that tend to form colloids. Moreover, factors such as coagulant dosage, pH value, water temperature, coexisting ions, etc., greatly influence the treatment effectiveness, making it challenging to control [11,12]. Membrane separation relies on the different pore sizes of membranes to separate dye molecules from water molecules, salts, etc., in dye wastewater, thereby enhancing wastewater reuse [13]. However, drawbacks such as membrane fouling, difficulty in membrane cleaning and the challenging disposal of concentrates limit its widespread use [14]. Adsorption involves using adsorbents with large surface areas and porous structures to adsorb dye wastewater, achieving wastewater treatment. Due to its low cost and simplicity, adsorption has been widely applied in wastewater treatment. Moreover, the repeated use of adsorbents after separation and recovery can improve the utilization rate of adsorbent materials, thus preventing secondary pollution and reducing adsorption costs [15]. Commonly used adsorbents include activated carbon [16], natural minerals [17], resin materials [18], nanomaterials [19], etc. Magnetic separation is a method that utilizes the different magnetic properties of various pollutants in dye wastewater, allowing the pollutants to undergo heterogeneous phase transfer under a magnetic field, thereby achieving separation and recovery. This technology is simple and efficient, but attention should be paid to thorough processing during operation to avoid secondary pollution [20,21].

TiO_2_ is an excellent adsorbent for anionic dye wastewater [22], but its high cost, low particle size, and tendency to agglomerate pose challenges. These characteristics make it difficult to achieve solid–liquid separation using conventional filtration methods after adsorption, leading to a loss of the adsorbent, difficulties in recovery, increased costs and secondary pollution issues [23]. Typically, this issue can be addressed by loading TiO_2_ onto a carrier material. Commonly used carriers include glass, zeolite, ceramics and kaolin. Shanxi, as the main production area of coal-series kaolin in China, possesses abundant resources of coal-series kaolin. Using it as a carrier not only reduces costs and pollution but also enables comprehensive resource utilization. Additionally, its large specific surface area and microporous structure could enhance the adsorption efficiency of the material. To achieve magnetic separation of the adsorbent material, coal-series kaolin can be modified and combined with γ-Fe_2_O_3_. The resulting γ-Fe_2_O_3_-modified coal-series kaolin is a cost-effective and highly magnetic adsorbent. After adsorption, it can be rapidly separated from the reaction system using magnetic separation technology. However, its limited adsorption capacity for anionic organic dyes restricts its application range. Combining TiO_2_ with γ-Fe_2_O_3_-modified coal-series kaolin produces a composite material that not only exhibits adsorption properties for anionic organic dyes but also enables magnetic separation and recovery. This combination proves to be mutually beneficial.

This study utilizes γ-Fe_2_O_3_-modified coal-series kaolin as a magnetic carrier and tetrabutyl titanate as a titanium source to prepare a TiO_2_/γ-Fe_2_O_3_-modified coal-series kaolin magnetic adsorbent regeneration material (hereinafter referred to as the composite material) using a co-precipitation method. Also, the study investigates its adsorption capacity for Congo red dye, aiming to provide new scientific basis for efficient and renewable functional composite adsorbent materials, thereby expanding their application range in environmental protection and achieving the goal of green governance.

## 2. Results and Discussion

### 2.1. Chemical Composition and Phase Composition of Raw Coal-Series Kaolin

#### 2.1.1. Chemical Composition

The chemical composition of the raw coal-series kaolin was analyzed using chemical analysis method, and the analysis results are shown in Table 1.

#### 2.1.2. Phase Composition

From Figure 1, it can be seen that the main minerals contained in the raw coal-series kaolin are kaolinite (JCPDS No.79-1570), quartz (No.89-1916), siderite (No.65-3107) and rutile (No.29-1360), etc.

### 2.2. XRD of Samples

Comparing Figure 2 (a), (b) and (c), it can be observed that the diffraction peaks of γ-Fe_2_O_3_-modified coal-series kaolin at 2θ = 31.1°, 35.5°, 54.8°, 63.0° correspond to the (2 2 0), (3 1 1), (4 2 2), (4 4 0) crystal planes of γ-Fe_2_O_3_ standard card JCPDS (No.65-3107), indicating the successful composite of TiO_2_ with γ-Fe_2_O_3_-modified coal-series kaolin.

Through the comparison of Figure 3a–c, it can be observed that the TiO_2_/γ-Fe_2_O_3_-modified coal-series kaolin composite material exhibits diffraction peaks at 2θ = 31.1°, 35.5° and 63.0°, corresponding to the (2 2 0), (3 1 1), and (4 4 0) crystal planes of γ-Fe_2_O_3_ standard card JCPDS (No.65-3107), respectively. The peaks at 2θ = 25.2°, 37.8° and 48.9° correspond to the (1 0 1), (0 0 4) and (2 0 0) crystal plane diffraction peaks of rutile-type TiO_2_ standard card (No.29-1360). During the synthesis process, there is no peak shift in the characteristic peak positions of TiO_2_/γ-Fe_2_O_3_-modified coal-series kaolin, indicating that the crystal structure remains unchanged after the composite of TiO_2_ with γ-Fe_2_O_3_-modified coal-series kaolin, demonstrating the successful composite formation.

### 2.3. N_2_ Adsorption-Desorption of Samples

As shown in Figure 4, the N_2_ adsorption–desorption isotherms of both exhibit Langmuir IV-type curves, with a significant H1-type hysteresis loop observed in Figure 4b between pressures of 0.45 and 0.95, indicating a relatively uniform pore size distribution in the composite material. The experimental results reveal that introducing a γ-Fe_2_O_3_/kaolin composite into TiO_2_ decreases the adsorption capacity. The TiO_2_/γ-Fe_2_O_3_-modified kaolin composite material has a specific surface area of 127.5 m^2^/g and a pore volume of 0.38 cm^3^/g, while TiO_2_ possess a specific surface area of 156.7 m^2^/g and a pore volume of 0.51 cm^3^/g. Compared to TiO_2_, both the specific surface area and pore volume of the composite material decrease due to the introduction of γ-Fe_2_O_3_-modified kaolin [24,25].

### 2.4. IR Spectrogram of the Composites

As can be seen from Figure 5a, γ-Fe_2_O_3_ has characteristic absorption peaks at 2964 cm^−1^, 1464 cm^−1^, 543 cm^−1^ and 471 cm^−1^. In Figure 5b, the characteristic absorption peaks of TiO_2_ can be seen at 2963 cm^−1^, 1463 cm^−1^ 500–700 cm^−1^. From the comparison of Figure 5a–c, it is not difficult to see that TiO_2_/γ-Fe_2_O_3_-modified coal-series kaolin obviously appears the characteristic peaks ofγ-Fe_2_O_3_ and TiO_2_, indicating that TiO_2_ successfully composes with γ-Fe_2_O_3_-modified coal-series kaolin.

### 2.5. SEM of the Composites

As shown in Figure 6, the raw coal-series kaolin (a) has a clear layered structure with relatively neat edges; the acid-modified coal-series kaolin (b) still has a layered structure, but it can be seen that the material’s structure becomes thinner, with burrs appearing at the edges, becoming loose and porous. This is due to the dissolution of Al_2_O_3_ during hydrochloric acid treatment of coal bearing kaolin, which leads to an increase in material pores and specific surface area; the surface of TiO_2_/γ-Fe_2_O_3_-modified coal-series kaolin (c) becomes relatively smooth and dense. This is due to the smaller particle sizes of Fe_2_O_3_ and TiO_2_, which fill the surface of modified coal-series kaolin to form a relatively uniform film layer. At the same time, the loading of TiO_2_ causes a large number of active hydroxyl groups on the surface of the composite, creating favorable conditions for adsorbing anionic organic compounds.

### 2.6. Magnetic Analysis and Magnetic Separation Experimental Results of the Composite Material

From Figure 7, it can be observed that the composite material exhibits a strong magnetism, with an Ms value of 13.4 emu/g, enabling solid–liquid phase separation using magnetic separation in the adsorption system. The results of magnetic separation testing using a magnet after static adsorption of Congo red by the composite material also demonstrate that the composite material is enriched on the side closer to the magnet under the influence of the magnetic field. This indicates the strong magnetic properties of the composite material and its feasibility for magnetic separation.

### 2.7. Single-Factor Adsorption Experiment of the Composite Material

#### 2.7.1. Influence of Adsorption Time

The influence of time on the adsorption of Congo red by the composite material is shown in Figure 8. From Figure 8, it can be observed that the adsorption rate of Congo red dye by the composite material reaches 96.8% within 10 min. As the adsorption time increases, the adsorption rate increases slowly with no significant change, reaching 98.8% at 100 min. The main reason for this phenomenon is that during the initial stage of adsorption, the composite material has a large number of effective adsorption sites on the surface, a high dye molecule concentration, a large effective mass transfer area and a strong mass transfer driving force, resulting in a high adsorption rate. As time progresses, the number of effective adsorption sites decreases, and the adsorption of dye molecules on the surface of the composite material is significantly enhanced due to the occupation effect. The adsorption reaches saturation and reaches the adsorption equilibrium point, maintaining a relatively stable level of adsorption rate. Therefore, considering the efficiency of adsorption, an adsorption time of 30 min is sufficient.

#### 2.7.2. The Influence of pH Value

The effect of pH value on the adsorption of Congo red by the composite material is shown in Figure 9. Figure 9 demonstrates that the rate of adsorption of Congo red dye by the composite material is significantly influenced by changes to the pH value. Overall, with an increase in pH, the adsorption rate initially increases, then decreases, followed by a sharp decline. When the pH is less than 5, the adsorption rate increases from 90.2% to 98.1% as the pH increases. This is because the surface of the carrier acid-modified coal-series kaolin in the composite material itself contains hydrogen ions, making it acidic (with a pH of approximately 3–4). When the solution is highly acidic, there is greater repulsion between them, which is not conducive to the adsorption of Congo red molecules onto the composite material. As the pH gradually increases, the Congo red surface forms anions, which combine with the cations on the surface of the composite material through electrostatic action, leading to a certain degree of increase in the adsorption rate. When the pH is greater than 5, the adsorption rate decreases slightly with increasing pH. This is mainly because the enhanced alkalinity of the solution hinders the protonation process on the surface of the composite material, weakening the electrostatic force between them and reducing the adsorption effect. When the pH is 12, a large number of negative charges accumulate on the surface of the composite material, causing it to lose its adsorption capacity for Congo red molecules almost completely, resulting in a sharp decrease in the adsorption rate to nearly zero. Thus, it can be seen that the pH value of the dye solution is an important factor affecting the adsorption performance of the composite material, with the optimal adsorption effect occurring at pH values between 5 and 9.

#### 2.7.3. The Influence of Temperature

The effect of temperature on the adsorption of Congo red by the composite material is shown in Figure 10. From Figure 10, it can be observed that the adsorption performance of the composite material is not greatly affected by temperature changes. With increasing temperature, the molecular kinetic energy increases, leading to a faster adsorption speed and higher adsorption rates. However, as the temperature continues to rise, the enhanced molecular thermal motion leads to an increase in desorbed Congo red molecules. Meanwhile, the increase in temperature may cause an internal structural expansion of the composite material, resulting in Congo red molecules detaching from the material surface, thereby causing a decrease in adsorption rate despite the temperature increase, reaching a saturation point. Therefore, adsorption by the composite material can be effectively carried out at room temperature.

In order to compare the adsorption capacity of the composite material, modified coal system kaolin andγ-Fe_2_O_3_-modified coal system kaolin to Congo red, an adsorption experiment on Congo red solution has been carried out in previous work. The adsorption results show that the adsorption rate of modified coal kaolin, γ-Fe_2_O_3_-modified kaolin reaches 98.4%, compared with the adsorption rate of composite materials at 10 min. However, the composite materials have the characteristics of anionic dyes and magnetism.

### 2.8. Adsorption Kinetics and Adsorption Thermodynamics Analysis

#### 2.8.1. Adsorption Kinetics Analysis

To analyze the adsorption kinetics characteristics of the composite material on Congo red, the experimental data were fitted into the first-order adsorption kinetics equation and pseudo-second-order adsorption kinetics equation [26]. In the Figure 11, the deviation coefficient R^2^ of the quasi second order kinetic model is 0.9980, and the kinetic rate constant k is 0.091 (g/mg·min), indicating that the quasi second order kinetic model is more suitable for describing the kinetic behavior of adsorption of Congo Red by adsorbent materials.

#### 2.8.2. Adsorption Thermodynamics Analysis

To analyze the thermodynamic characteristics of the adsorption of Congo red onto the composite material, the experimental data were separately fitted into the Langmuir isotherm adsorption equation and the Freundlich isotherm adsorption equation for linear fitting, resulting in Figure 12 and Figure 13. The results indicate that the adsorption basically conforms to both the Langmuir and Freundlich isotherm adsorption equations. However, as shown in Figure 12 and Figure 13, in the Freundlich model, the deviation coefficient R^2^ = 0.8832 indicates insufficient fit, while in the Langmuir model, it is 0.9664, which is more reliable. The adsorption of Congo red molecules on the surface of the adsorbent material is monolayer adsorption, with a theoretical maximum adsorption capacity of 78.5 mg/g.

### 2.9. Regeneration of Composite Materials

As can be seen from Figure 14, the composite material has a better adsorption effect on Congo red after three regenerations, with average adsorption rates of 94%, 92% and 91%, respectively. This indicates that EDTA has a superior desorption effect on the composite material, and also suggests that the composite material possess excellent regenerative performance and can be reused.

## 3. Materials and Methods

### 3.1. Materials

Tetrabutyl titanate, acetylacetone, both of chemically pure, were purchased from Shanghai National Pharmaceutical Group Chemical Reagent Co., Ltd. (Shanghai, China). Other reagents including Congo red, hydrochloric acid, EDTA, sodium hydroxide, concentrated ammonia solution, concentrated nitric acid, iron nitrate, anhydrous ethanol and ethanol, all analytically pure, were purchased from Tianjin Chemical Reagent Factory (Tianin, China).

The coal-series kaolin was obtained from a coal mine in Datong, Shanxi Province.

### 3.2. Instrumentation

Major instruments included UV-Vis 2550 UV-visible Spectrophotometer (Shimadzu, Kyoto, Japan), Nova 4000E Specific Surface and Porosity Analyzer (Quantachrome, Tallahassee, FL, USA), Ultimal IV X-ray Diffractometer (Japan Science Corporation, Rigaku Ultima IV, Tokyo, Japan) and 7407 Vibrating Sample Magnetometer (Lake Shore, WV, USA), NEXUS 607 Fourier transformation infrared spectrometer (Nicolet, Madison, WI, USA), SU8010 Scanning electron microscope (Hitachi, Tokyo, Japan).

### 3.3. Methods

#### 3.3.1. Composition and Phase Analysis of Coal-Series Kaolin

The chemical composition of coal-series kaolin samples was analyzed using chemical analysis methods to determine their chemical composition. X-ray diffraction was used to characterize the samples and determine their phase composition.

#### 3.3.2. Acid Modification of Coal-Series Kaolin

Calcined coal-series kaolin was obtained by calcining the coal-series kaolin at 600 °C for 8 h, followed by surface activation treatment with 6 mol/L hydrochloric acid.

#### 3.3.3. Preparation and Characterization of γ-Fe_2_O_3_-Modified Coal-Series Kaolin

We placed 2 g of modified coal-series kaolin in a conical flask, dispersed and it with anhydrous ethanol as a dispersant under ultrasonic dispersion for 30 min, followed by the addition of 10 g of iron nitrate and concentrated ammonia solution under stirring. After continuing stirring for 3 h, all substances were washed with anhydrous ethanol, filtered, and air-dried at room temperature. They were then placed in an oven and calcined at 180 °C for 2 h. After grinding and sieving, magnetic γ-Fe_2_O_3_-modified coal-series kaolin was obtained. X-ray diffraction was used to characterize the γ-Fe_2_O_3_-modified coal-series kaolin and other samples.

#### 3.3.4. Preparation of TiO_2_

Add 0.5 mL of acetylacetone to 30 mL of anhydrous ethanol, add 8.5 mL of tetrabutyl titanate, stir for 30 min and let old for 24 h, then put the resulting sol into the oven, dry at 100 °C for 5 h, grind in a mortar to powder, and then put in mav furnace at 500 °C for calcination for 5 h to obtain TiO_2_ powder.

#### 3.3.5. Preparation and Characterization of Composite Materials

We mixed 30 mL of ethanol with 0.5 mL of acetone, followed by the addition of 8.5 mL of tetrabutyl titanate. After stirring for 10 min, solution A was obtained. γ-Fe_2_O_3_-modified coal-series kaolin was added to solution A. After stirring and dispersing for 1 h, solution B, containing 0.45 mL of distilled water, 0.7 mL of concentrated nitric acid and 4.5 mL of ethanol, was added. The mixture was stirred for 2 h, aged for 24 h, then filtered, dried, calcined, cooled, ground and sieved to prepare TiO_2_/γ-Fe_2_O_3_-modified coal-series kaolin composite materials. X-ray diffraction, specific surface and porosity analyzer, vibrating sample magnetometer, Fourier transformation infrared spectrometer and Scanning electron microscope were used for characterization of the composite materials.

Using this preparation method, the mass percentage content of TiO_2_ is 50%, and the best value obtained from previous experiments is used; that is, the ratio of TiO_2_ to γ-Fe_2_O_3_-modified coal system kaolin mass is 1:1.

#### 3.3.6. Magnetic Separation Experiment of Composite Materials

A certain amount of composite material was placed in a 250 mL conical flask, and 20 mL of 100 mg/L Congo red solution was added. After vibrating for a period, the mixture was allowed to stand, and then magnetic separation was performed on the adsorbed solution using a magnet.

#### 3.3.7. Single-Factor Adsorption Experiment of Composite Materials

To investigate the adsorption factors of the composite materials, 20 mL of 100 mg/L Congo red solution was placed in a 250 mL conical flask, and a certain amount of composite material was added under certain conditions for vibrational adsorption.

The concentration of Congo Red in the mixture was measured by spectrophotometry at its maximum absorption wavelength of 497 nm, and then calculated using the Lambert Beer formula.
*A* = *Kbc*(1)
where *A* is the absorbance, *K* is the molar absorption coefficient, *b* is the thickness of the liquid layer (cm), *c* is the concentration of the solution (g/L).

This experiment studied the adsorption effect of composite materials on Congo red under three different conditions: time, pH value and temperature, and the adsorption rate of composite materials on Congo red was calculated using the following formula:(2)$$\theta =\frac{C_{0}-C_{T}}{C_{0}}\times 100\%$$
where *θ*, *C*_0_, *C_T_* represent the adsorption rate, the mass concentration of Congo red in dye wastewater before adsorption (mg/L) and the mass concentration of Congo red in dye wastewater after adsorption equilibrium (mg/L), respectively.

The isothermal adsorption model refers to the change curve for the concentration relationship between solute molecules in the solution and adsorbent phases when reaching adsorption equilibrium at a certain temperature, which macroscopically reflects the adsorption characteristics and is of great significance for the use of adsorbents. The Langmuir and Freundlich isotherm adsorption equations are the most classic among the common adsorption isotherm models. The Langmuir adsorption equation has special application conditions: the adsorption is uniform single-layer adsorption, and the adsorption heat is constant. The solute is completely independent and the adsorption equilibrium is dynamic equilibrium. The Freundlich equation has no assumptions and is applicable to adsorption in most cases [27].

The Langmuir Equation (3) and Freundlich Equation (4) are as follows:(3)$$\frac{C_{e}}{q_{e}}=\frac{1}{q_{m}K_{L}}+[\frac{1}{q_{m}}]C_{e}$$
(4)$$\log q_{e}=\log K_{F}+\frac{1}{n}\log C_{e}$$

*q_m_* is the theoretical maximum adsorption capacity (mg/g); *K_L_* is the Langmuir isotherm constant (L/mg), *KF* (mg/g); and 1/*n* is the Freundlich constant. The larger the *K_F_* and *n* values, the better the adsorption performance.

The commonly used adsorption kinetics models are the quasi first-order kinetics model (5) and the quasi second-order kinetics model (6), and their formulas are expressed as follows:(5)$$\log\left(q_{e}-q_{t}\right)=\log q_{e}-\frac{k_{1}t}{2.303}$$
(6)$$\frac{t}{q_{t}}=\frac{1}{k_{2}q_{e}^{2}}+\frac{t}{q_{e}}$$
where *t* is the time (min), *q_t_* is the adsorption amount at time *t* (mg/g) and *k*_1_ and *k*_2_ are the rate constants of the quasi first and quasi second order adsorption kinetic models, respectively.

#### 3.3.8. Regeneration Experiment of Composite Materials

After adsorption, an appropriate amount of 0.1 mol/L EDTA solution was added to the adsorbed composite material, followed by vibration desorption. After approximately 60 min, filtration, washing and drying were carried out to obtain the regenerated adsorbent. Then, the regenerated adsorbent was used to adsorb Congo red dye wastewater. This process was repeated 3 times for regeneration, with 3 repeated adsorptions after regeneration.

## 4. Conclusions

The TiO_2_/γ-Fe_2_O_3_-modified coal-series kaolin composite material prepared in this study has a tested specific surface area of 127.5 m^2^/g, a pore volume of 0.38 cm^3^/g and an Ms value of 13.4 emu/g. The composite material shows a fast adsorption rate for Congo red dye at 100 mg/L, reaching 96.8% within 10 min, following a pseudo-second-order kinetic model. The adsorption process conforms to the Langmuir IV-type monolayer adsorption. The pH of the Congo red dye promotes a significant amount of adsorption before reaching 11. After three regeneration cycles, the rate of adsorption of Congo red by the composite material remains above 91%. In summary, the composite material exhibits good adsorption and magnetic properties, making it suitable for use as an adsorbent for anionic organic dyes. After adsorption, solid–liquid phase magnetic separation can be achieved, demonstrating its regenerative and reusable characteristics.

## Figures and Tables

**Figure 1 molecules-29-02964-f001:**
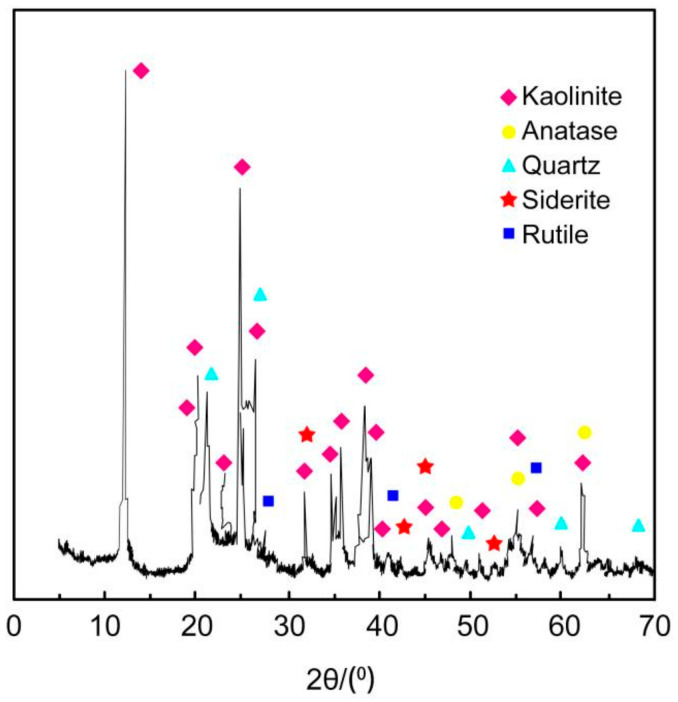
The XRD pattern of the raw coal-series kaolin.

**Figure 2 molecules-29-02964-f002:**
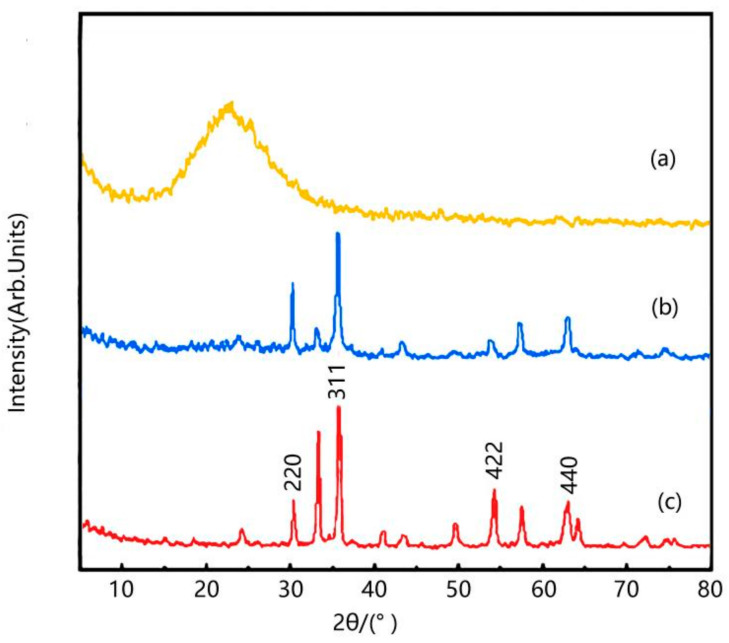
XRD pattern of sample. (**a**) Modified coal-series kaolin; (**b**) γ-Fe_2_O_3_/modified coal-series kaolin; (**c**) γ-Fe_2_O_3_.

**Figure 3 molecules-29-02964-f003:**
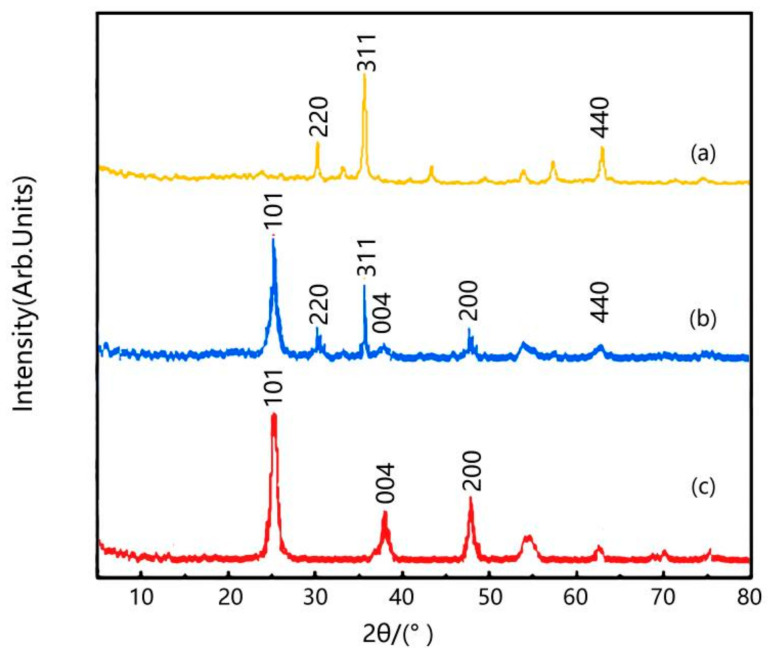
XRD pattern of sample. (**a**) γ-Fe_2_O_3_-modified coal-series kaolin; (**b**) TiO_2_/γ-Fe_2_O_3_-modified coal-series kaolin; (**c**) TiO_2_.

**Figure 4 molecules-29-02964-f004:**
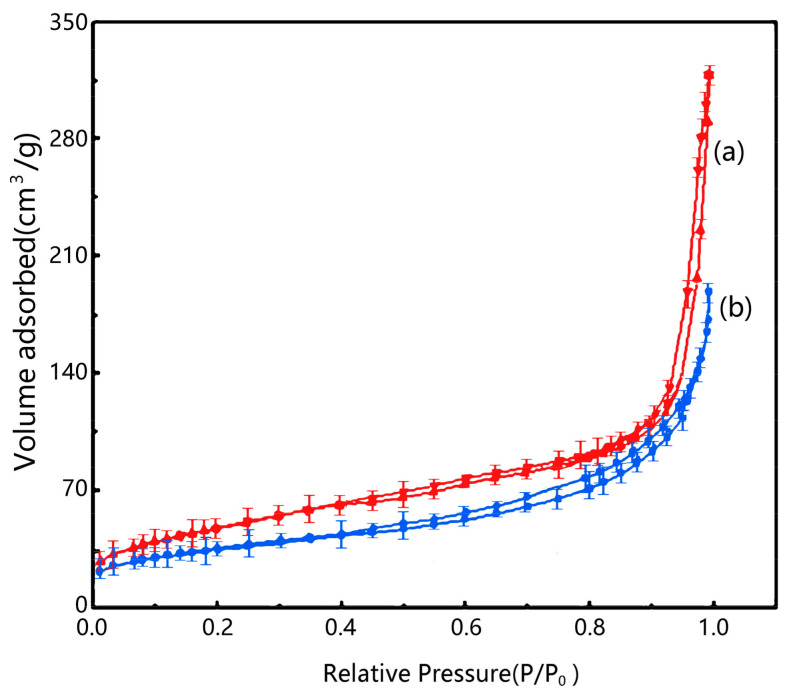
N_2_ adsorption–desorption isotherm of the sample. (**a**) TiO_2_; (**b**) TiO_2_/γ-Fe_2_O_3_-modified coal-series kaolin.

**Figure 5 molecules-29-02964-f005:**
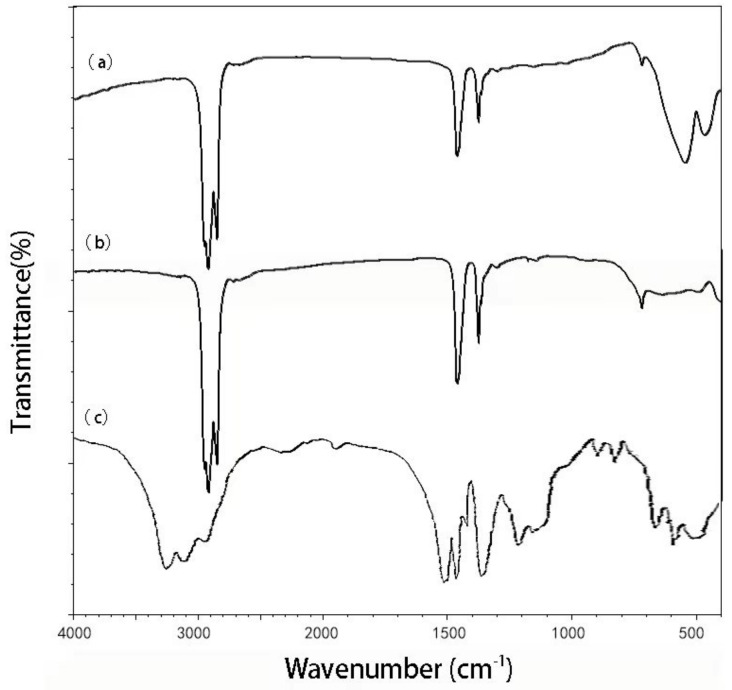
FTIR spectra of the of the samples. (**a**) Fe_2_O_3_ (**b**) TiO_2_; (**c**) TiO_2_/γ-Fe_2_O_3_-modified coal-series kaolin.

**Figure 6 molecules-29-02964-f006:**
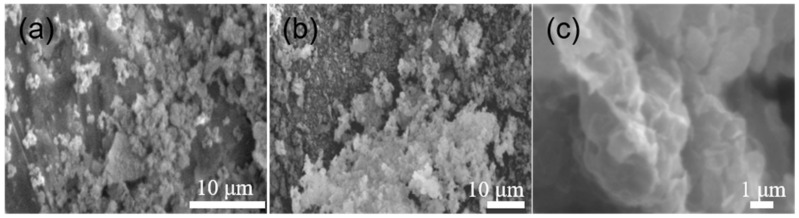
SEM of the samples. (**a**) Raw coal-series kaolin; (**b**) modified coal-series kaolin; (**c**) TiO_2_/γ-Fe_2_O_3_-modified coal-series kaolin.

**Figure 7 molecules-29-02964-f007:**
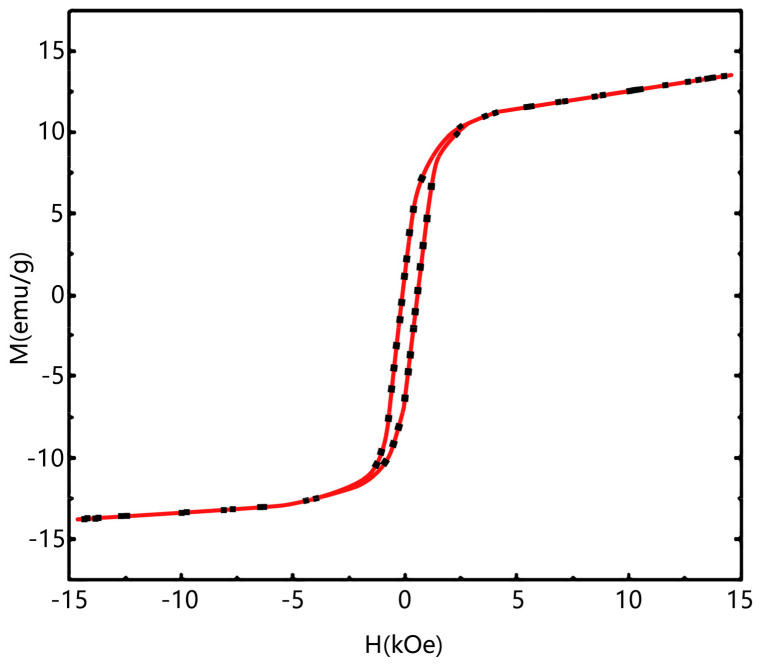
Hysteresis loop diagram of composite materials.

**Figure 8 molecules-29-02964-f008:**
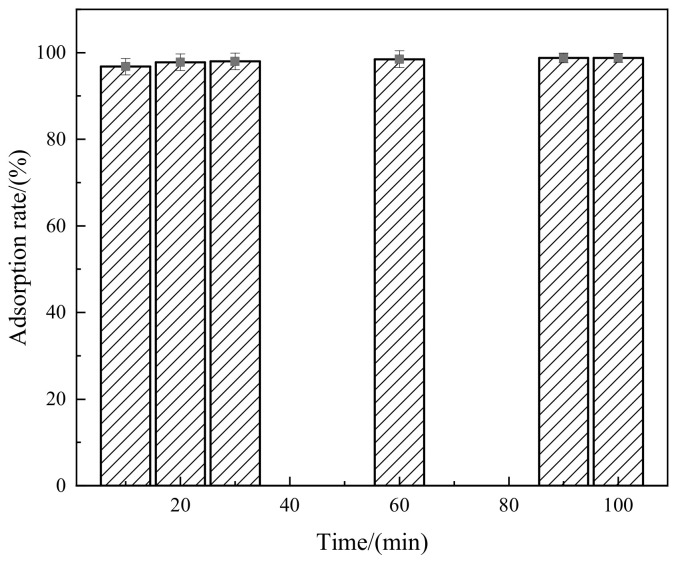
Relationship between adsorption time and adsorption rate.

**Figure 9 molecules-29-02964-f009:**
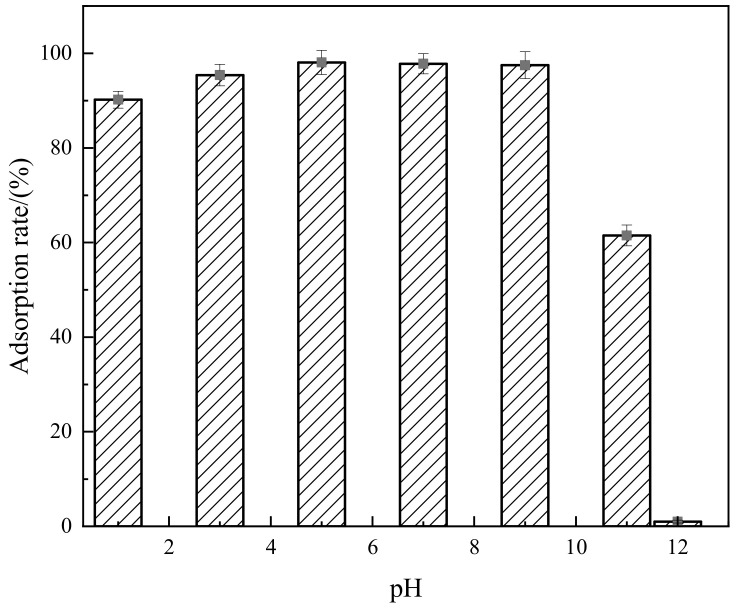
Relationship between pH value and adsorption rate.

**Figure 10 molecules-29-02964-f010:**
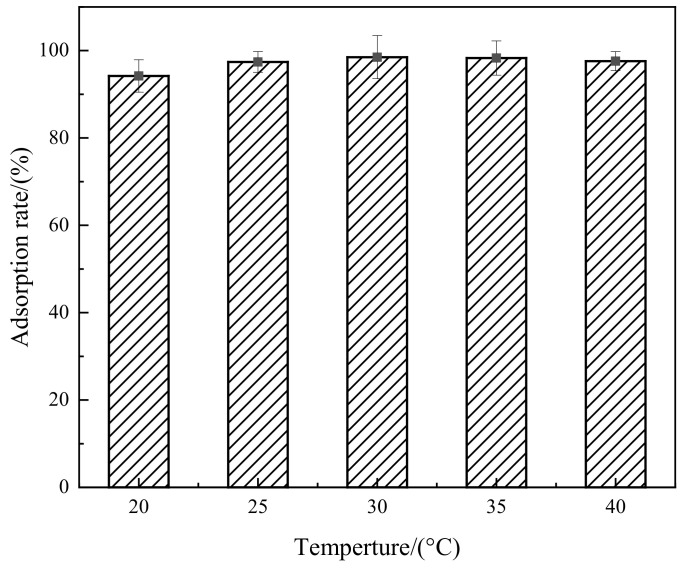
Relationship between temperature and adsorption rate.

**Figure 11 molecules-29-02964-f011:**
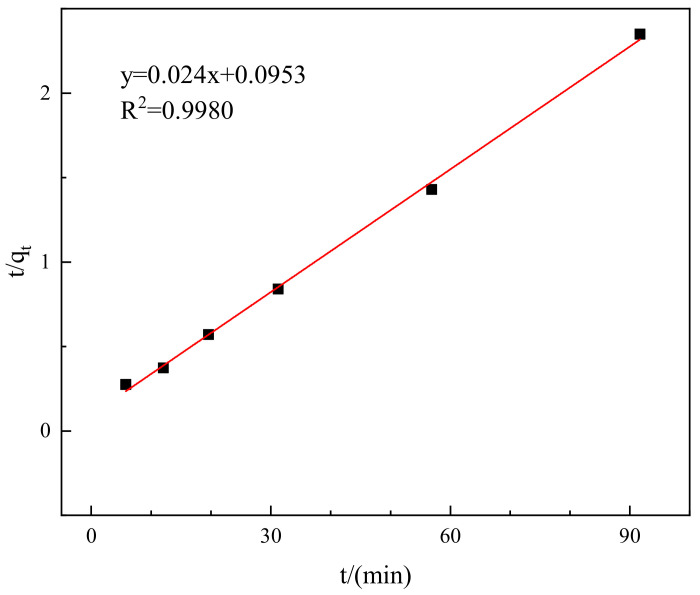
Pseudo-secondary adsorption kinetic model.

**Figure 12 molecules-29-02964-f012:**
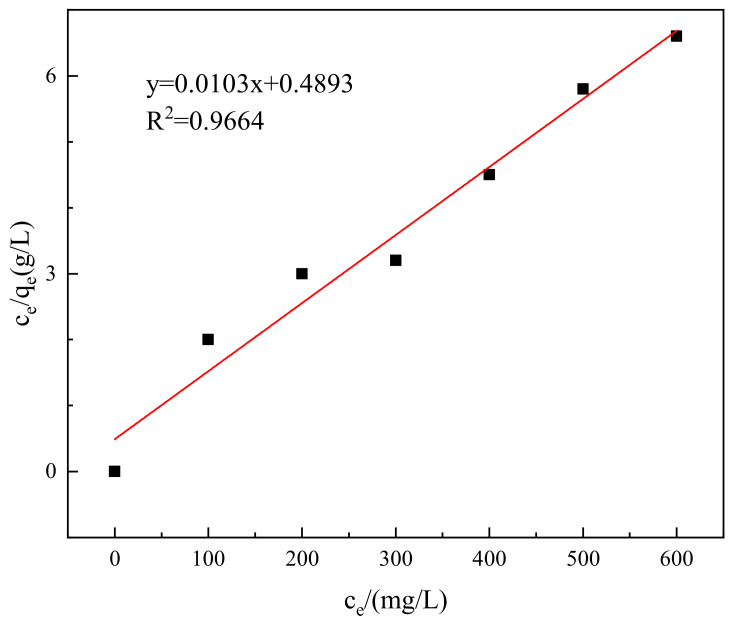
Langmuir adsorption isotherm.

**Figure 13 molecules-29-02964-f013:**
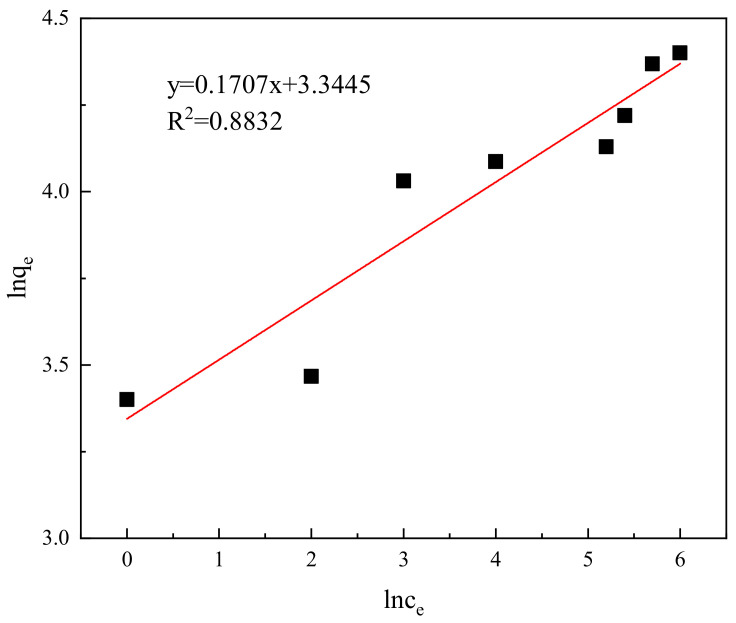
Freundlich adsorption isotherm.

**Figure 14 molecules-29-02964-f014:**
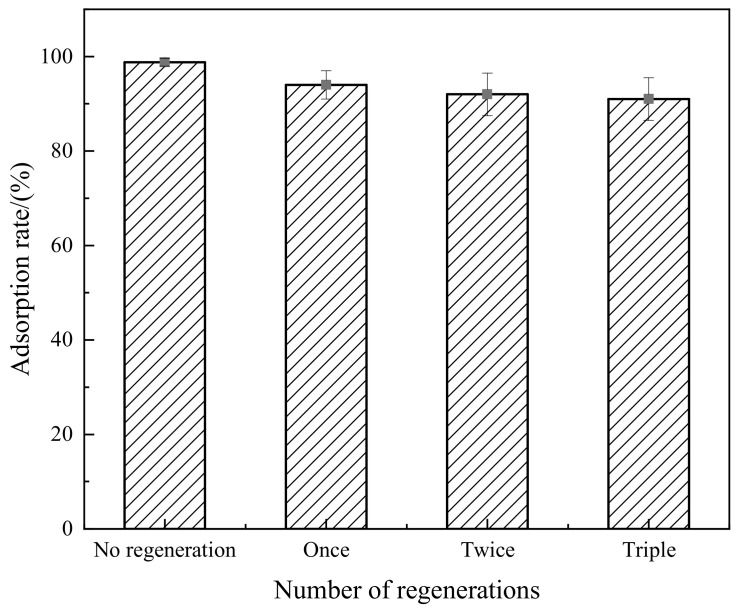
Relationship between regeneration times and adsorption rate.

**Table 1 molecules-29-02964-t001:** The chemical composition of raw coal-series kaolin.

Chemical Components	SiO_2_	Al_2_O_3_	Fe_2_O_3_	TiO_2_	K_2_O	CaO	Na_2_O	MgO	Others	Loss on Ignition
content/%	45.42	37.52	0.48	0.63	0.14	0.42	0.02	0.15	1.42	13.8

## Data Availability

No new data were created or analyzed in this study. Data are contained within the article.

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
