# Peer review of "Study of the Preparation and Performance of TiO2-Based Magnetic Regenerative Adsorbent"

_molecules, 2024, doi:10.3390/molecules29132964_

Round 1
Reviewer 1 Report
Comments and Suggestions for Authors
The manuscript presents an analysis of a regenerative adsorbent; however, there are several areas that require attention to enhance clarity and completeness.
In some of the figures, error bars are missing (figure 4), and experimental points are not consistently represented (igure 5). Including error bars and all experimental points would improve the accuracy and reliability of the data presentation, ensuring transparency in the analysis.
The discussion section, particularly regarding adsorption kinetics and adsorption thermodynamics analysis, lacks sufficient depth. Key equations essential for understanding these concepts are missing, which limits the readers' ability to fully grasp the theoretical framework underlying the analysis. Providing the relevant equations and expanding on the discussion of adsorption kinetics and thermodynamics would strengthen the manuscript's analytical rigor and theoretical grounding.
Overall, while the manuscript offers valuable insights into regenerative adsorbent, addressing these issues would enhance the clarity and completeness of the analysis, ensuring the manuscript meets the standards expected for publication in the field.
Author Response
Q1. In some of the figures, error bars are missing (figure 4), and experimental points are not consistently represented (figure 5). Including error bars and all experimental points would improve the accuracy and reliability of the data presentation, ensuring transparency in the analysis.
A1. Thanks for reviewer’s good comments. According to reviewer’s suggestion, we have added the error bar and experimental points in the revised manuscript.
Figure 4. N2 adsorption-desorption isotherm of the sample. (a) TiO2; (b) TiO2/γ-Fe2O3-modified coal-series kaolin.
Figure 5. Hysteresis loop diagram of composite materials.
Q2. The discussion section, particularly regarding adsorption kinetics and adsorption thermodynamics analysis, lacks sufficient depth. Key equations essential for understanding these concepts are missing, which limits the readers' ability to fully grasp the theoretical framework underlying the analysis. Providing the relevant equations and expanding on the discussion of adsorption kinetics and thermodynamics would strengthen the manuscript's analytical rigor and theoretical grounding.
A2. Thanks for raising this good point. Analysis on adsorption kinetics and thermodynamics was added as follows which can been seen in line 302-326:
The isothermal adsorption model refers to the change curve of the concentration relationship between solute molecules in the solution and adsorbent phases when reaching adsorption equilibrium at a certain temperature, which macroscopically reflects the adsorption characteristics and is of great significance for the use of adsorbents. The Langmuir and Freundlich isotherm adsorption equations are the most classic among the common adsorption isotherm models. The Langmuir adsorption equation has special application conditions: the adsorption is uniform single-layer adsorption, and the adsorption heat is constant; The solute is completely independent and the adsorption equilibrium is dynamic equilibrium. The Freundlich equation has no assumptions and is applicable to adsorption in most cases.
The Langmuir equation (2) and Freundlich equation (3) are as follows:
(2)
(3)
qm is the theoretical maximum adsorption capacity (mg/g); KL is the Langmuir isotherm constant (L/mg), KF (mg/g), 1/n is the Freundlich constant, The larger the KF and n values, the better the adsorption performance.
Dynamic model
The commonly used adsorption kinetics models are the quasi first order kinetics model (4) and the quasi second order kinetics model (5), and their formulas are expressed as follows:
(4)
(5)
t is the time (min),qt is the adsorption amount at time t (mg/g); k1 and k2 are the rate constants of the quasi first and quasi second order adsorption kinetic models, respectively.
And the analysis on adsorption kinetics and thermodynamics were explained in line 234-238, 248-252, as follows:
In the figure 10, the deviation coefficient R2 of the quasi second order kinetic model is 0.9980, and the kinetic rate constant k is 0.091 (g/mg.min), indicating that the quasi second order kinetic model is more suitable for describing the kinetic behavior of adsorption of Congo Red by adsorbent materials.
As shown in the figure 11 and 12, in the Freundlich model, the deviation coefficient R2=0.8832 indicates insufficient fit, while in the Langmuir model, it is 0.9664, which is more reliable. The adsorption of Congo red molecules on the surface of the adsorbent material is monolayer adsorption, with a theoretical maximum adsorption capacity of 78.5 mg/g.

Reviewer 2 Report
Comments and Suggestions for Authors
Review for the Manuscript «Preparation and performance study of TiO2-based magnetic regenerative adsorbent»
Authors Dahui Tian, Jiarui Hao, Xiaojie Guo submitted to the journal «Molecules»
My recommendation for this manuscript is - Reject.
And there are some points why:
1. Lack of characterization (any microscopy or spectroscopy?). Only XRD that could be used for the identification of the presence of Fe and Ti on the structure.
2. Adsorption thermodynamics analysis should be improved, what formulas were used? Isotherm parameters should be calculated and presented as well as the maximum equilibrium adsorption capacity.
3. No comparison of the proposed material with other methods widely presented in the literature for Congo red dye.
4. P. 3.2 have you mentioned reagents or instruments? Please specify.
5. If you are mentioned «green governance» you should explain the poin. The sintesys includes the use of acetone, concentrated nitric acid, and hydrochloric acid, concentrated ammonia solution. Please use the tools to calculate the «GREEnness» of the method.
Author Response
Q1. Lack of characterization (any microscopy or spectroscopy?). Only XRD that could be used for the identification of the presence of Fe and Ti on the structure.
A1. Thanks for the reviewer’s comment and suggestion. The infrared spectrum was added in the manuscript revised which can be seen as follows:
Line 283-284: NEXUS 607 Fourier transformation infrared spectrometer (Nicolet, USA).
line 315: and Fourier transformation infrared spectrometer were used for characterization of the composite materials.
Line 140:
2.4 IR spectrogram of the composites
As can be seen from Figure 5 (a), γ-Fe2O3 has characteristic absorption peaks at 2,964 cm-1,1,464 cm-1,543cm-1, and 471cm-1. In figure 5 (b), the characteristic absorption peaks of TiO2 were at 2963cm-1,1463cm-1,500-700cm-1. From the comparison of Figure (c), (a) and (b), it is not difficult to see that TiO2 /γ-Fe2O3-modified coal-series kaolin obviously appears the characteristic peaks of γ-Fe2O3 and TiO2, indicating that TiO2 successfully composes with γ-Fe2O3-modified coal-series kaolin.
Figure 5. FTIR spectra of the of the samples.(a) Fe2O3 (b) TiO2; (c) TiO2/γ-Fe2O3-modified coal-series kaolin.
Q2. Adsorption thermodynamics analysis should be improved, what formulas were used? Isotherm parameters should be calculated and presented as well as the maximum equilibrium adsorption capacity.
A2. Thanks for the reviewer’s question.
Analysis on adsorption kinetics and thermodynamics was added as follows which can been seen in line 302-326:
The isothermal adsorption model refers to the change curve of the concentration relationship between solute molecules in the solution and adsorbent phases when reaching adsorption equilibrium at a certain temperature, which macroscopically reflects the adsorption characteristics and is of great significance for the use of adsorbents. The Langmuir and Freundlich isotherm adsorption equations are the most classic among the common adsorption isotherm models. The Langmuir adsorption equation has special application conditions: the adsorption is uniform single-layer adsorption, and the adsorption heat is constant; The solute is completely independent and the adsorption equilibrium is dynamic equilibrium. The Freundlich equation has no assumptions and is applicable to adsorption in most cases.
The Langmuir equation (2) and Freundlich equation (3) are as follows:
(2)
(3)
qm is the theoretical maximum adsorption capacity (mg/g); KL is the Langmuir isotherm constant (L/mg), KF (mg/g), 1/n is the Freundlich constant, The larger the KF and n values, the better the adsorption performance.
Dynamic model
The commonly used adsorption kinetics models are the quasi first order kinetics model (4) and the quasi second order kinetics model (5), and their formulas are expressed as follows:
(4)
(5)
t is the time (min),qt is the adsorption amount at time t (mg/g); k1 and k2 are the rate constants of the quasi first and quasi second order adsorption kinetic models, respectively.
And the analysis on adsorption kinetics and thermodynamics were explained in line 234-238, 248-252, as follows:
In the figure 10, the deviation coefficient R2 of the quasi second order kinetic model is 0.9980, and the kinetic rate constant k is 0.091 (g/mg.min), indicating that the quasi second order kinetic model is more suitable for describing the kinetic behavior of adsorption of Congo Red by adsorbent materials.
As shown in the figure 11 and 12, in the Freundlich model, the deviation coefficient R2=0.8832 indicates insufficient fit, while in the Langmuir model, it is 0.9664, which is more reliable. The adsorption of Congo red molecules on the surface of the adsorbent material is monolayer adsorption, with a theoretical maximum adsorption capacity of 78.5 mg/g.
Q3. No comparison of the proposed material with other methods widely presented in the literature for Congo red dye.
A3. Thanks for reviewer’s comments. The proposed material with other methods were widely used such as activated carbon, natural minerals, resin materials, nanomaterials, etc. Due to its low cost and simplicity, adsorption has been widely applied in wastewater treatment. Moreover, the repeated use of adsorbents after separation and recovery can improve the utilization rate of adsorbent materials, thus preventing secondary pollution and reducing adsorption costs.
Mouni et al. (in the References of 17) conducted a systematic study on wastewater containing methylene blue dye using unmodified kaolin. The absorption characteristics of kaolin was studied through a thermodynamic model, and it was found that kaolin can absorb the maximum adsorption capacity at pH 6.0 and 25 ° C is 52.76 mg/g, indicating strong adsorption capacity. Meanwhile, the materiel can lead secondary pollution and reducing adsorption costs.
Q4. P. 3.2 have you mentioned reagents or instruments? Please specify.
A4. Thanks for reviewer’s comments. Sorry for the typos. We have deleted "reagents "in P.3.2.
Q5. If you are mentioned «green governance» you should explain the poin. The sintesys includes the use of acetone, concentrated nitric acid, and hydrochloric acid, concentrated ammonia solution. Please use the tools to calculate the «GREEnness» of the method.
A5. Thanks for reviewer’s comments. The “green governance”mentioned in the article refers to the green concept of using solid waste to treat wastewater and treating waste with waste. But as you said, the concept of green synthesis and green chemistry also needs to be considered in the synthesis process in order to achieve more perfect green governance. In future research, I will use the tools you mentioned to calculate and provide effective basis and methods for subsequent experiments.

Reviewer 3 Report
Comments and Suggestions for Authors
This manuscript provides a preparation of TiO2/Fe2O3-kaolin composite, its characterization and study of its adsorption properties towards Congo Red dye. The experimental results showed high adsorption degree of the dye and efficient magnetic separation of the adsorbent.
In my opinion, this work needs to be improved to meet the requirements of the journal Molecules. Some comments may be helpful:
1. line 81: what is the role of titanium sulfate mentioned in this paragraph? Please, indicate its formation or addition in the experimental procedure?
2. What is the method used to follow the concentration of Congo Red in the reaction mixture? Please, explain it to the readers.
3. Figure 3 (c) shows the XRD of TiO2. However, no information about the sample. Is it TiO2 standard or TiO2 prepared by the same synthetic method?
4. The authors claimed that TiO2 is an excellent adsorbent. Related to that, a question arises about the percentage of TiO2 in the composite. Please, provide data.
5. Additional experiment is required to show the adsorption properties of support – kaolin as well as Fe2O3-kaolin to evaluate the effect of TiO2.
6. line 298: Have the authors checked the concentration of Congo Red in EDTA solution? What was the utilization of EDTA solution?
Comments on the Quality of English Language
The text should be checked carefully.
Author Response
Q1. line 81: what is the role of titanium sulfate mentioned in this paragraph? Please, indicate its formation or addition in the experimental procedure?
A1. Thanks for reviewer’s good comments. According to reviewer’s suggestion, The titanium sulfate in line 81 should be changed to tetrabutyl titanate, which was used as Titanium source to synthetize TiO2/Fe2O3 kaolin composite materials.
Q2. What is the method used to follow the concentration of Congo Red in the reaction mixture? Please, explain it to the readers.
A2. Thanks for the reviewer’s comment. Add this sentence on line of line 303: The concentration of Congo Red in the mixture was measured by spectrophotometry at its maximum absorption wavelength of 497nm, and then calculated using the Lambert Beer formula.
A=Kbc
where A is the absorbance, K is the molar absorption coefficient, b is the thickness of the liquid layer (cm), c is the concentration of the solution (g/L)
Q3. Figure 3 (c) shows the XRD of TiO2. However, no information about the sample. Is it TiO2 standard or TiO2 prepared by the same synthetic method?
A3. Thanks for the reviewer’s comment and suggestion. The sample of TiO2 is prepared by the same synthetic method. The method is added in line 301 which can be seen in the manuscript revised.
3.3.4 Preparation of TiO2
Add 0.5 ml of acetylacetone to 30 ml of anhydrous ethanol, add 8.5 ml of tetrabutyl titanate, stir for 30 minutes and let old for 24 hours, then put the resulting sol into the oven, dry at 100 0C for 5 hours, grind in a mortar to powder, and then put in mav furnace 500 0C calcination for 5 hours to get TiO2 powder.
Q4. The authors claimed that TiO2 is an excellent adsorbent. Related to that, a question arises about the percentage of TiO2 in the composite. Please, provide data.
A4. Thanks for the reviewer’s comment and suggestion. The percentage of TiO2 in the composite was tested about 1:1 which was added in line 316.
In this preparation method, the mass percentage content of TiO2 is 50%, and the best value obtained from previous experiments is used, that is, the ratio of TiO2 to γ -Fe2O3-modified coal system kaolin mass is 1:1.
Q5. Additional experiment is required to show the adsorption properties of support – kaolin as well as Fe2O3-kaolin to evaluate the effect of TiO2.
A5. Thanks for the reviewer’s comment and suggestion. In order to compare the adsorption capacity of composite material, modified coal system kaolin andγ-Fe2O3-modified coal system kaolin to Congo red, the adsorption experiment to Congo red solution has been done in the previous work. The adsorption results show that the adsorption rate of modified coal kaolin, γ-Fe2O3-modified kaolin reaches 98.4%, compared with the adsorption rate of composite materials at 10min. However, the composite materials have the characteristics of anionic dyes and magnetism.
Q6. line 298: Have the authors checked the concentration of Congo Red in EDTA solution? What was the utilization of EDTA solution?
A6. Thanks for the reviewer’s comment and suggestion. Considering the main determination of the adsorption performance of the regenerated adsorbent material, the concentration of Congo Red in EDTA solution was not measured; The main function of EDTA solution is desorption.
Round 2
Reviewer 1 Report
Comments and Suggestions for Authors
In figure 5 (now 6) (hysteresis loop) I suggest to change experimental point colour from red to black
Author Response
Q1. In figure 5 (now 6) (hysteresis loop) I suggest to change experimental point colour from red to black.
A1. Thanks for reviewer’s good comments. According to reviewer’s suggestion, we have changed the colour red to black.

Reviewer 2 Report
Comments and Suggestions for Authors
Dear Authors,
XRD and IR are not enough to characterize the material and prove the structure.
Author Response
Q1. XRD and IR are not enough to characterize the material and prove the structure.
A1. Thanks for the reviewer’s comment and suggestion. The SEM was added in the manuscript revised which can be seen as follows:
Figure 6. SEM of the samples.(a) Raw coal-series kaolin (b) Modified coal-series kaolin
(c) TiO2/γ-Fe2O3-modified coal-series kaolin.

Reviewer 3 Report
Comments and Suggestions for Authors
The authors followed my comments. Thank you.
Author Response
thanks